# Recent Advances in C–H Functionalization of Pyrenes

Srinivasarao Arulananda Babu *, Arup Dalal and Subhankar Bodak

Department of Chemical Sciences, Indian Institute of Science Education and Research (IISER) Mohali, Knowledge City, Sector 81, SAS Nagar, Mohali 140306, Punjab, India
* Correspondence: sababu@iisermohali.ac.in

**Abstract:** In recent years, transition metal-catalyzed C–H activation and site-selective functionalization have been considered to be valuable synthetic tactics to functionalize organic compounds containing multiple C–H bonds. Pyrene is one of the privileged and notorious polycyclic aromatic hydrocarbons. Pyrene and its derivatives have found applications in various branches of chemical sciences, including organic chemistry, chemical biology, supramolecular sciences, and material sciences. Given the importance of pyrene derivatives, several classical methods, including the C–H functionalization method, have been developed for synthesizing modified pyrene scaffolds. This review attempts to cover the recent developments in the area pertaining to the modification of the pyrene motif through the C–H activation process and the functionalization of C–H bonds present in the pyrene motif, leading to functionalized pyrenes.

**Keywords:** C–H activation; C–H functionalization; directing group; polycyclic aromatic hydrocarbon; pyrene; synthetic methods





## 1. Introduction

Pyrene and its derivatives have gained significant interest in various branches of chemical sciences, including organic chemistry, chemical biology, supramolecular sciences, and material sciences [1–25]. The exceptional photophysical properties of pyrene derivatives, coupled with their capability to facilitate efficient energy transfer, have led to significant investigation and innovation. Photophysical properties, including exceptional fluorescence characteristics, effective excimer emission, and impressive charge carrier mobility, attribute the position of pyrene as an important element for constructing a diverse array of small molecules and materials. Pyrene is a pivotal building block in the fabrication of applied materials, including organic light-emitting diodes (OLEDs), organic semiconductors used in organic field-effect transistors (OFETs), supramolecular sensors, solar cells, etc. (e.g., compounds **1a**–**g**, Figure 1). Further, the versatility of pyrene-based organic materials has led to their exploitation in various categories of photoelectric devices [1–10]. Furthermore, the pyrene motif has been admired for its binding capabilities, and it can be involved in π-stacking and C–H–π interactions. This property has been widely utilized in the non-covalent modification of various extended planar π-systems, such as carbon nanotubes and graphene sheets [11–18]. Additionally, pyrene derivatives have found applications in the field of chemical biology, e.g., in constructing systems to bind to nucleic acids [4] and in creating artificial receptors for aromatic and carbohydrate molecules [19–25].

The specific substituents incorporated and their corresponding positions within the pyrene molecule **2a** have been found to influence the optoelectronic and photophysical characteristics of pyrene derivatives [1–25]. The 1-, 3-, 6-, and 8-positions of the pyrene molecule identified are referred to as 'active' or 'common sites' (Figure 2) [9,10]. These sites possess a higher electron density and readily undergo electrophilic aromatic substitution (SEAr) reactions. The synthesis of functionalized pyrene derivatives by introducing substitutions at these sites has been commonly explored. On the other hand, the 2- and 7-positions of pyrene are designated as 'nodal plane positions' and are considered 'uncommon' or 'less

accessible sites for functionalization'. Thus, functionalizing the 2- and 7-positions of pyrene is considered challenging. The other positions, 4-, 5-, 9-, and 10-, are called K-regions due to the carcinogenic effect of pyrene upon its oxidation [1–10].

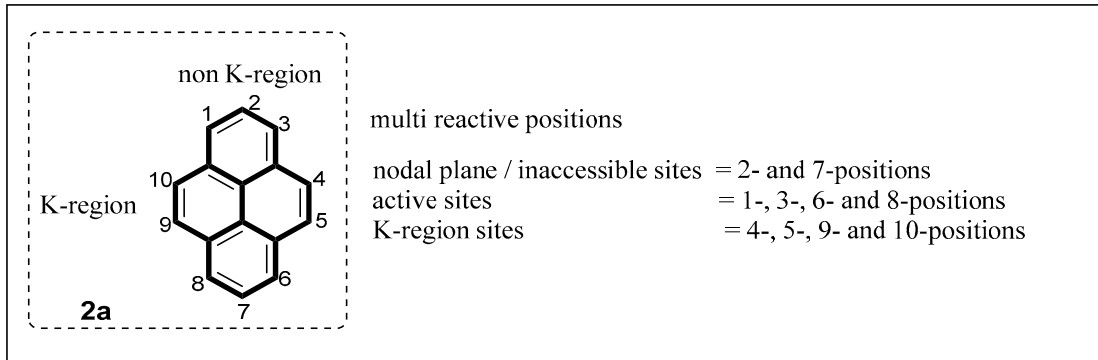

**Figure 1.** Representative examples of pyrene-based materials **1a**–**g**.

**Figure 2.** Multi-reactive position of the pyrene molecule.

The substantial orbital co-efficient present at positions 1-, 3-, 6-, and 8- enables the occurrence of electrophilic reactions at these carbon atoms [1–10]. Representative primitive works related to the substitution of pyrene at different sites are described in Scheme 1 [10]. Vollmann et al. demonstrated the preliminary synthesis of pyrene derivatives bearing mono-, bis-, tri-, or tetra-substituted groups [26]. Introducing substituents at the 2- and 7-positions in a single step was considered difficult, and multi-step pathways were established to synthesize 2- and 7-disubstituted pyrenes [9,10]. In 2005, Marder et al. [27] reported the iridium-catalyzed direct C–H borylation of pyrene at the 2- and 7-positions, affording the corresponding products **3e** and **3f**. An intermediate complex [Ir(bpy)(Bpin)$_3$] formed in the process seemed to play a key role in selectively engaging the C–H bonds present at the 2- and 7-positions (Scheme 1). Notably, the C(2)- and C(7)-borylated pyrene derivatives have been utilized for synthesizing various pyrene derivatives possessing alcohol, ether, triflate, and bromide functionalities and also new pyrene products formed through classical methods [10,28–33] and cross-coupling reactions [34]. The treatment of

pyrene with bromine has led to bromination at the 1-, 3-, 6-, and 8-positions, affording the corresponding products **3a–d** [35]. The reaction with bromine [9,10] does not lead to the functionalization of the four positions, such as 4-, 5-, 9-, and 10- in the K-region of pyrene [35]. Similarly, the introduction of bulky *tert*-butyl groups at positions 2 and 7 does not effectively alter the bromination outcome by preventing access to the active positions 1, 3, 6, and 8 [9,10,36–39]. Pyrenes containing the bromide substituent at different positions were used in cross-coupling reactions to obtain functionalized pyrenes [9,10]. Friedel–Crafts alkylation with *tert*-butyl chloride in the presence of AlCl$_3$ gave 2,7-di-*tert*-butyl pyrene [40]. Notably, the installation of the *tert*-butyl group served as the bulky substituent to introduce other groups at the different sites in the pyrene core [10]. Oxidation reactions at the 4-, 5-, 9-, and 10-positions of the K-region of pyrene have been carried out, which led to the assembling of **3g,h** (Scheme 1), and their oxidation chemistry was also explored to obtain various functionalized pyrenes [10,41–43].

(i): Br$_2$
(ii): [Ir(OMe)COD]$_2$ (5 mol%), dtbpy (10 mol%), B$_2$pin$_2$, cyclohexane, 80 °C, 16 h.
(iii): RuCl$_3$, NaIO$_4$, DCM-CH$_3$CN-H$_2$O (1:1:1.25), rt or 40 °C.

**Scheme 1.** Representative primitive works related to the substitution of pyrenes at different sites [27,35,41–43].

Given the notorious applications of pyrene and its derivatives, developing valuable pyrene-based materials with tuned molecular structures and photoelectric properties would be possible by introducing different functional group attachments into the suitable positions of the pyrene core. While the substitution of pyrenes in all positions would be valuable, apart from the functionalization of the active positions, a few methods enabling selective and effective functionalization of some of the inaccessible sites of pyrene suffer from limitations.

In recent years, C–H functionalization has emerged as one of the most fruitful strategies to introduce a functional group at the C–H bonds of the small molecules, and is considered an alternative method to the cross-coupling reactions [44–64]. Particularly, the transition metal-catalyzed C–H bond activation and functionalization of various classes of organic molecules has been well documented. C–H activation and functionalization can be accomplished in two ways: (i) via direct C–H activation/functionalization; and (ii) via directing group-assisted C–H activation/functionalization. C–H activation and functionalization depend on the reactivity of substrates, metal catalysts, and other conditions. The

direct C–H activation/functionalization of a molecule does not provide the desired product with efficiency, as there will be regio- or site-selectivity issues. Notably, the directing group-assisted C–H activation concept has helped to overcome the regioselectivity issues when similar types of C–H bonds are present in a molecule. The directing group initially coordinates to a transitional metal and brings the coordinating metal center to a C–H bond present in proximity in a molecule, enabling site selection [54–64]. There have been various efforts to use the functional groups present in organic molecules as directing groups. Notably, the transition metal-catalyzed bidentate directing group-assisted C–H functionalization of small organic molecules, viz., carboxamides possessing or assembled using a bidentate directing group, has been considered a reliable method [54–64]. Accordingly, the Pd(II)-catalyzed bidentate directing group-aided functionalization including C–H arylation, alkylation, halogenation, amidation, and oxygenation, are well-documented methods in organic synthesis. Different directing groups, such as 8-aminoquinoline, 2-(methylthio)aniline, 4-amino-2,1,3-benzothiadiazole, 2-picolinic acid, etc., have been introduced to accomplish the C–H functionalization of molecules at desired positions [44–64]. A wide range of organic compounds, including aromatic, heteroaromatic, heterocyclic, aliphatic, and alicyclic compounds, along with amino acids, have been subjected to site-selective C–H functionalization [54–64].

Nevertheless, the pyrene motif was also subjected to C–H activation and functionalization to obtain substituted pyrenes [10]. This review was focused on delineating the recent developments in the area pertaining to the modification of the pyrene skeleton involving the transition metal-catalyzed functionalization of C–H bonds of the pyrene skeleton, affording functionalized pyrenes. This review covered the current status of various C–H functionalization reactions, including arylation, alkylation, olefination, etherification, chalcogenation, allylation, and carbonylation of the pyrene core at different sites (Scheme 2).

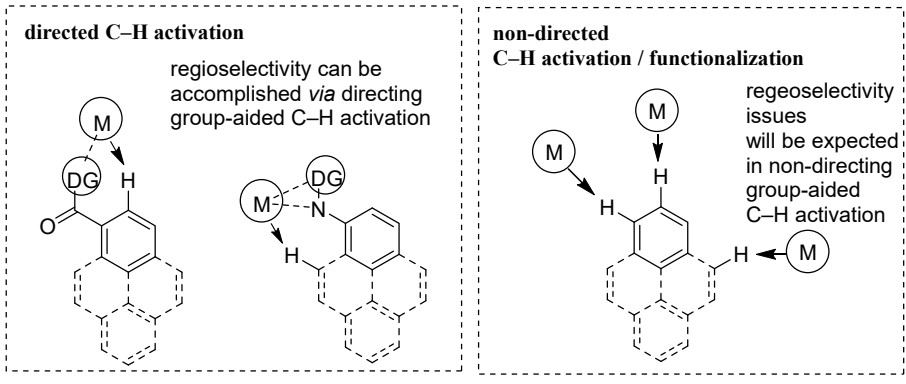

DG = directing group
M = metal

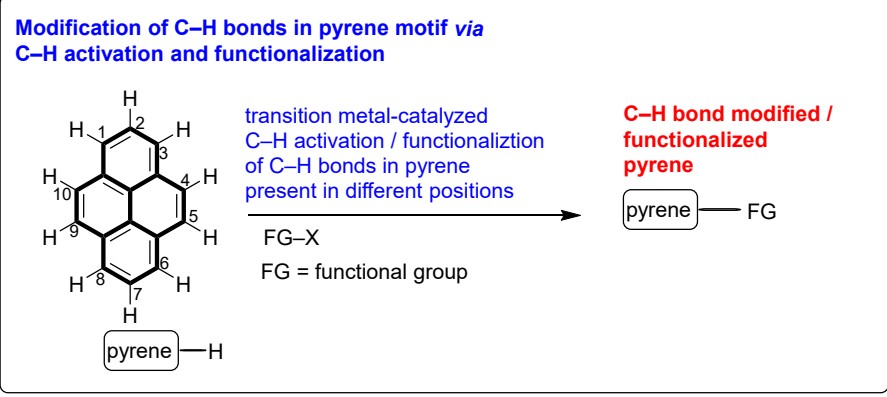

**Scheme 2.** C–H activation and functionalization of organic molecules.

## 2. Direct C–H Activation and Functionalization of Pyrene, Affording Functionalized Pyrenes

In this section, we have presented the recent developments pertaining to the direct C–H activation and functionalization of pyrene motifs without using any directing group. In 1993, Perutz et al. reported [65] one of the primitive C–H activation attempts of the pyrene core and the synthesis of an arene–rhodium complex through C–H activation of pyrene, naphthalene, perylene, and triphenylene via C–H bond cleavage. The rhodium catalyst (**4a**) successfully formed an $\eta^2$-complex with pyrene, while naphthalene, perylene, and triphenylene gave an equilibrium mixture of their corresponding $\eta^2$-complex and C–H activation products. Pyrene was added into a solution of **4a** in hexane and heating the reaction mixture at 60 °C for 21 h gave the pyrene-based $\eta^2$-complex **4b** with a 90% yield (Scheme 3).

**Scheme 3.** Attempt at C–H activation in pyrene and synthesis of the pyrene-based $\eta^2$-complex [65].

### 2.1. Direct Functionalization of the C2 and C7 Positions of Pyrenes

Marder et al. described an Ir-catalyzed [27]-selective borylation of polycyclic aromatic hydrocarbons, including pyrene and the structure of pyrene-2,7-bis(boronate) esters. Borylation reactions occurred at the C(2) and C(7)-positions and resulted in the formation of the pyrene-2,7-bis(boronate) ester. The catalyst derived in situ between [Ir(OMe)COD]$_2$ and 4,4′-di-*tert*-butyl-2,2′-bipyridine (dtbpy) demonstrated notable effectiveness and specificity in facilitating the C–H borylation reaction of pyrene [27,66]. The observed site selectivity was attributed to the highly congested structure of the five-coordinate *fac*-tris(boryl)species, designated as [Ir(bpy)(Bpin)$_3$]. This complex has been believed to be a crucial intermediate, which is responsible for the pivotal C–H activation step that determines the reaction rate. The reaction of pyrene (1 equiv), B$_2$pin$_2$ (1.1 equiv), [Ir(OMe)COD]$_2$ (5 mol%), and dtbpy (10 mol%) in cyclohexane at 80 °C for 16 h produced the C(2)-borylated product **3e** and the C(2), C(7)-bis-borylated product **3f** of 68% and 6% yields, respectively. The exclusive synthesis of the C(2), C(7)-bis-borylated product (97%) was also achieved using excess boron reagent (2.2 equiv of B$_2$pin$_2$) (Scheme 4).

| B$_2$pin$_2$ | **3e** | **3f** |
|---|---|---|
| 1.1 equiv | 68% | 6% |
| 2.2 equiv | - | 97% |

**Scheme 4.** Functionalization of the C2 and C7 positions of pyrenes. Synthesis of borylated pyrenes via C–H borylation [27,66].

Driess designed [67] a cobalt catalyst termed **5**, which has been used for the C–H borylation reaction of pyrene core **2a**, affording the C(2), C(7)-bis-borylated pyrene **3f**. Catalyst

**5** (8 mol%), NaBHEt$_3$ (16 mol%), and 2 equiv of B$_2$Pin$_2$ in the presence of cyclohexene (2 equiv) resulted in the bis-borylated product of pyrene **3f** with a 78% yield (Scheme 5). A tentative mechanistic pathway for the Co-catalyzed C–H borylation of pyrene was proposed, involving cobalt(I) hydride generated in situ upon the addition of NaHBEt$_3$ to the Co catalyst **5**. Then, B$_2$Pin$_2$ undergoes oxidative addition to cobalt(I) hydride, forming an intermediate, which, after reductive elimination of HBPin, produces an active cobalt(I) boryl intermediate. Then, a C–H oxidative addition reaction of pyrene to the cobalt(I) boryl intermediate, followed by reductive elimination, results in the expected C–H-borylated product **3f**. When cyclohexene is available, it interacts with cobalt(I) hydride via a coordination insertion pathway, and this intermediate with HB results in cobalt(I) hydride and cobalt(I) boryl intermediates, respectively. It was suggested that the elevated levels of HBPin in the catalytic cycle help to generate cobalt(I) boryl intermediates in the absence of cyclohexene.

**Scheme 5.** Functionalization of the C2 and C7 positions of pyrenes. Synthesis of borylated pyrenes via C–H borylation [67].

Murai's group presented [68] an iridium-catalyzed intermolecular dehydrogenative silylation reaction of polycyclic aromatic compounds, including pyrene **2a**, without using directing groups (Scheme 6). The C2-substituted **6a1** pyrene as well as the C2- and C7-disubstituted **6a2** pyrene were obtained. Pyrene, HSiEt$_3$ (0.5 equiv), [Ir(OMe)(COD)]$_2$ (2.5 mol%), tmphen (5 mol%), and 3,3-dimethyl-1-butene (2 equiv) were mixed in 1,4-dioxane and heated for 9 h at 100 °C to afford the C2-silylated product **6a1** (54%) and C2, C7-silylated product **6a2** (3%). In another reaction, pyrene was treated with 3 equiv of HSiEt$_3$ in the presence of [Ir(OMe)(COD)]$_2$ (5 mol%), tmphen (10 mol%), and 3,3-dimethyl-1-butene (3 equiv) in 1,4-dioxane at 100 °C for 9 h. This reaction gave the C2-silylated product **6a1** (63%) and C2, C7-silylated product **6a2** (20%). The authors stated a plausible mechanism for the intermolecular dehydrogenative silylation reaction. An Ir–H species is generated through the oxidative addition reaction of Ir–OMe to Et$_3$SiH, followed by a reductive elimination reaction, releasing Et$_3$Si–OMe. Then, the initially generated Ir-H species reacts with Et$_3$SiH, undergoes insertion into 3,3-dimethyl-1-butene to afford **7c** (via **7b**), and subsequently undergoes reductive elimination to afford Ir-SiEt$_3$ species. Ir–SiEt$_3$ then undergoes oxidative addition to the C(sp$^2$)–H bond of pyrene **2a**, affording intermediate **7d**, which then undergoes reductive elimination to yield the C2-substituted pyrene **6a1**. The C2 silylation of **2a** followed by C7 borylation afforded **6b**. Subsequently, C2- and C7-disubstituted pyrene **6b** was used as a coupling partner in the cross-coupling reaction, and it was subjected to the sequential cross-coupling reactions to afford the pyrene motif **9a**.

## 2.2. Direct Functionalization of the C4 Position of Pyrenes

Following the earlier work on direct C–H borylation of C2 and C7 positions of pyrene, Liu and Marder's group revealed [66] the direct C–H borylation of pyrene core **2a** at the C(4) position using [Ir(COD)Cl]$_2$/dtbpy as the catalyst precursor, while the C(2)- and C(7)-positions of pyrene **5b** are occupied with a bulky substituent. Accordingly, the C–H-borylated pyrene **10b** was obtained, and then it was subjected to the cross-coupling reaction to afford **11a** (Scheme 7). Along this line, the C(2), C(7)-bis-borylated pyrene **3f**

was obtained from pyrene **2a**. Then, **3f** was subjected to C(4)–H borylation to afford the tris C—-H-borylated product **10c**. Subsequently, **10c** was subjected to the cross-coupling reaction to afford **11b** (Scheme 7).

**Scheme 6.** Functionalization of the C2 and C7 positions of pyrenes. Iridium-catalyzed intermolecular dehydrogenative silylation of pyrenes and plausible mechanism for the silylation of pyrenes [68].

(i): *t*-BuCl, AlCl$_3$, DCM, rt, 16 h
(ii) B$_2$pin$_2$, [Ir(COD)Cl]$_2$, dtbpy, hexane, 80 °C, 48 h
(iii) 4-bromotoluene, Pd(PPh$_3$)$_4$, Na$_2$CO$_3$, toluene/H$_2$O, 80 °C, 16 h

**Scheme 7.** Functionalization of the C4 position of pyrenes. Synthesis of borylated pyrenes via C–H borylation [66].

Itami reported [69,70] the direct C–H activation and arylation of pyrenes **2a** and **10a** with arylboroxins **12a** as arylating agents using Pd(OAc)$_2$ as the catalyst and *o*-chloral as an oxidant, affording C–H-arylated pyrene **13**. Accordingly, a wide range of pyrenes, namely **13a–i**, **13j–l**, **14a,b**, and **14c**, were assembled (Scheme 8). The screening of the reaction conditions via varying oxidants and solvents resulted in the mixture of Pd(OAc)$_2$ (2.5 mol%) and *o*-chloral (1 equiv) in DCE at 80 °C as the optimized condition for accomplishing the synthesis of the C–H-arylated pyrene **13** via direct C–H arylation of pyrene. Other oxidants, such as DDQ, *p*-chloranil, *p*-benzoquinone, CuCl$_2$, and K$_2$S$_2$O$_8$, were ineffective. Other than *o*-chloranil, 2-benzoquinone was found to display some reactivity, affording the C–H arylation product. The C4 selectivity that was observed could be explained via various pathways. One possible mechanism proposed by Itami et al. [69] was based on the electrophilic palladation of pyrene with aryl palladium species at the C1 position, forming a cationic intermediate species, which then generates intermediates (**15c** via **15a,b**) through the σ-π-σ isomerization process. Subsequent deprotonation of intermediates (**15c**) followed by reductive elimination generates the C–H-arylated product **13** via **15d** (Scheme 9). Alternatively, the formation of a palladium complex at the C4–C5 double bond and the electrophilic palladation reaction would generate a cationic intermediate (**15c**). Furthermore, the Heck-type insertion of aryl palladium species may take place to form an intermediate (**15f** via **15e**), which then gives the C–H-arylated product **13** either via β-hydrogen elimination or via protodepalladation (through **15g**). Furthermore, Itami et al. demonstrated [70] the formation of 4-mesitylpyrene when pyrene was treated with Pd(OAc)$_2$ (10 mol%), *o*-chloranil (1 equiv), and AgOTf (20 mol%) in mesitylene (36 equiv) at 50 °C for 14 h. Along this line, C4-arylated pyrene was synthesized by Glorious et al. using Pd/C (Scheme 8) [71] Two different coupling partners were employed to accomplish

the C–H arylation reaction of pyrene **2a** to obtain their corresponding C4-arylated pyrenes **13b,c,g**. Pyrene **2a** was treated with either [Ar$_2$I]BF$_4$ (1 equiv) or [ArI(TRIP)]BF$_4$ in the presence of Pd/C (5 mol%) in DME at 100 °C to give the desired C–H-arylated pyrenes. In this case, the reaction seemed to be involving a similar plausible mechanism proposed by Itami et al., affording the C–H-arylated pyrenes (Schemes 8 and 9) [69].

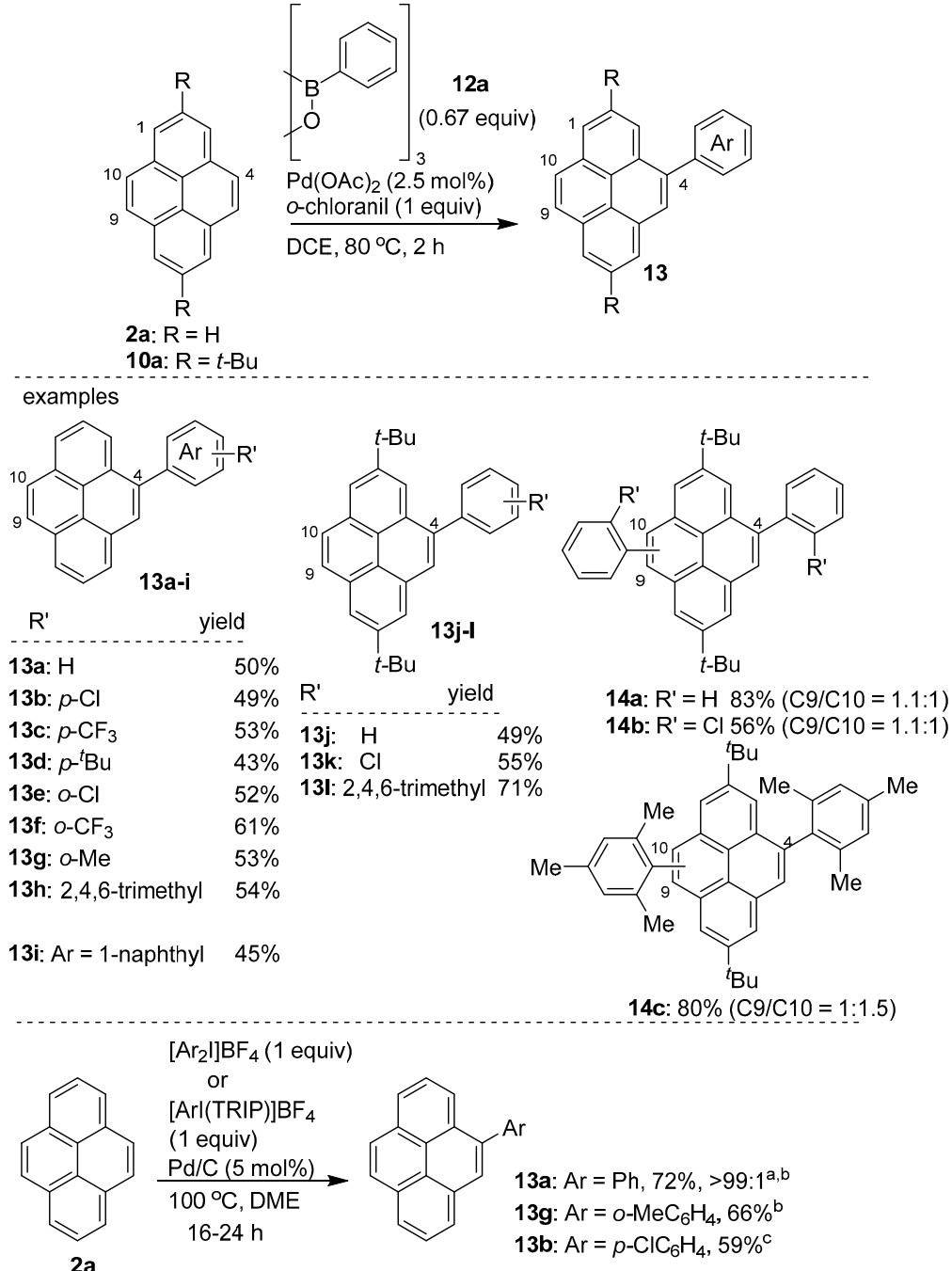

**Scheme 8.** Functionalization of the C4 and C10 positions of pyrenes. Synthesis of C–H-arylated pyrenes via direct C–H arylation of pyrenes [69–71].

**Scheme 9.** A plausible mechanism proposed by Itami et al. for the synthesis of C–H-arylated pyrenes via direct C–H arylation of pyrenes at the C4 position [69–71].

Oi et al. demonstrated [72] a palladium-catalyzed C–H bond arylation reaction of arenes, including pyrene with aryltrimethylsilanes, in the presence of $CuCl_2$ as an oxidant (Scheme 10). Pyrene, phenyltrimethylsilane (2 equiv), $PdCl_2$ (5 mol%), and $CuCl_2$ (4 equiv) in 1,2-DCE were heated for 16 h at 80 °C to afford **13a** (34%) and **16a** (14%). Electrophilic transmetalation of $PdCl_2$ with $ArSiMe_3$ takes place at the *ipso*-Si position, generating an aryl palladium intermediate in the presence of $CuCl_2$. Subsequently, aromatic electrophilic substitution occurs between pyrene **2a** and the aryl palladium intermediate, leading to the formation of a diaryl palladium intermediate (**16b**). The exact structure of the aryl palladium intermediate has not been definitively determined, but the involvement of $CuCl_2$ in this step is considered crucial for the reaction to proceed. Reductive elimination takes place from the diaryl palladium species **16b**, affording the cross-coupled product **13**. Then, $CuCl_2$ oxidizes the palladium species to regenerate $PdCl_2$.

### 2.3. Direct Functionalization of the C1 and C6 Positions of Pyrenes

Agarwal et al. established [73] a ferrocene-catalyzed C–H arylation reaction of arenes including pyrene, and a reaction mechanism study using cyclic voltammetry. Ferrocene-catalyzed C–H arylation of pyrene was accomplished using an aryldiazonium salt as an arylating agent (Scheme 11). The formation of arylated pyrene **13a** was observed by treating pyrene **2a** and phenyl diazonium tetrafluoroborate **17a** in the presence of ferrocene (10 mol%) at rt in acetone medium. The formation of a radical intermediate in the initial step of the reaction was evident by the EPR experiment, which was further supported by the DFT calculation. A plausible reaction mechanism was stated. Ferrocene triggers the formation of a phenyl radical (**17d**) from **17a** via the single-electron transfer mechanism and **17c**. The formation of phenyl radicals was investigated via cyclic voltammetry. Then, the phenyl radical reacts with pyrene **2a**, followed by electron transfer to a ferrocenium ion (via **17e**), and abstraction of a proton from **17f** generates C–H-arylated pyrene **13a**.

**Scheme 10.** Functionalization of the C4 position of pyrenes. Palladium-catalyzed C–H bond arylation of pyrenes with aryltrimethylsilane in the presence of $CuCl_2$ [72].

Yoshida et al. divulged [74] an example of metal- and chemical-oxidant-free electrochemical C–H/C–H coupling of indole **17h** and pyrene **2a** compounds using radical cation pools (Scheme 11). Anodic oxidation was carried out using an H-type divided cell equipped with a carbon felt anode and a platinum plate cathode. The electrochemical C–H/C–H coupling of indole and pyrene compounds was performed using the following typical procedure to afford **17j**. An indole compound solution (0.66 mmol) in a 0.1 M solution of $Bu_4NPF_6$ in DCM (10 mL) was placed in the anodic chamber. Then, trifluoromethanesulfonic acid (150 mL) and a 0.1 M solution of $Bu_4NPF_6$ in DCM (10 mL) were placed in the cathodic chamber. Constant current electrolysis (8.0 mA) was carried out at −78 °C with magnetic stirring of the reaction mixture for 60 min. Pyrene substrate (0.05 mmol) and 1,2-dimethoxyethane (0.1 mL) were added to the anodic chamber at −40 °C. The resulting mixture was stirred at −40 °C for 3 h, followed by at −15 °C, and the reaction mixture was stirred for 1.5 h. Then, $Et_3N$ (0.2 mL) was added, and the resulting mixture was warmed to room temperature and subjected to workup and purification to afford the product **17j**.

König's group revealed [75] the C–H phosphonylation reaction of electron-rich arenes and heteroarenes using visible light photoredox catalysis. Pyrene **2a** was reacted with $Ru(bpz)_3$ (2 mol%), $(NH_4)_2S_2O_8$, and $P(OEt)_3$ in $CH_3CN$ at 25 °C for 20 h under visible light irradiation (455 nm). The reaction yielded two products, namely C1-substituted pyrene (**18a**) and C1- and C6-disubstituted pyrene (**18b**), both with a 48% yield (Scheme 12). A plausible mechanism was stated by the authors of [75] for the C–H phosphonylation reaction of electron-rich arenes, such as pyrene **2a** using visible light photoredox catalysis. Upon photoexcitation, the photocatalyst $[Ru^{II}(bpz)_3][PF_6]_2$ produces an excited state $[*Ru^{II}(bpz)_3][PF_6]_2$, which accepts an electron from pyrene **2a**, converting it into its corresponding pyrenyl cation. Ammonium persulfate $(NH_4)_2S_2O_8$ accepts an electron from the reduced $Ru^{1+}$ species, completing the catalytic cycle and affording a sulfate dianion and a sulfate radical anion. The pyrenyl radical cation reacts with nucleophilic $P(OEt)_3$, generating the species **18c**. Hydrogen atom abstraction via $SO_4{}^-$ anions leads to the generation of a transient pyrenyl phosphonium intermediate species. Then, with the help of the sulfate dianion, the desired C–H phosphonylation product is generated. Takai et al. disclosed [76] the nitration of pyrene under mild conditions using $Fe(NO_3)_3 \cdot 9H_2O$, which

generates a nitrogen dioxide free radical species, giving product **19a** with a 89% yield. This C–H nitration process does not need any acidic promoters, pyridine-based-directing groups, ionic liquids, or supported metal nitrates (Scheme 12). Itami's group reported [77] a gold-catalyzed C–H imidation reaction of polycyclic aromatic hydrocarbons and the introduction of an amino group at the C(2) position of the pyrene core **2a/2aa** (Scheme 12). In the presence of AuCl (10 mol%) and biquinoline **19b** (12 mol%), the pyrene core reacted with NFSI in DCE at 70 °C for 12 h. The formation of a Au–ligand complex is believed to react with NFSI to generate an imidyl radical, which would then undergo radical addition to the pyrene core. Single-electron transfer from the resulting radical intermediate affords a cationic intermediate, which is then aromatized to generate the corresponding imidated pyrene **19c**.

**Scheme 11.** Functionalization of the C1 position of pyrenes. Ferrocene catalyzed C–H arylation of pyrenes using an aryldiazonium salt as an arylating agent and electrochemical C–H/C–H coupling of indole and pyrene compounds using radical cation pools [73,74].

**Scheme 12.** Functionalization of the C1 and C6 positions of pyrenes. Direct C–H phosphonylation of pyrenes via visible light photoredox catalysis, C–H nitration, and amination of the pyrene core [75–77].

### 2.4. Direct Functionalization of the C9 and C10 Positions of Pyrenes

Nowicka and Willock disclosed [78] their experimental and computational approaches comprising the mechanistic insights of the selective ruthenium ion-catalyzed oxidation of pyrenes. A mechanistic investigation into the ruthenium ion-catalyzed oxidation of various aromatic hydrocarbons led to an understanding of the chemistry of aromatic C=C bond cleavage. The DFT calculations showed that the regioselectivity in the reaction can be understood in terms of the preservation of aromaticity in the initial formation of a metallocycle at the C(9), C(10) and C(4), C(5) double bonds of pyrene **2a** (Scheme 13). Two competing pathways comprising the C=C bond cleavage leading to a dialdehyde **20h** and C–H activation followed by H-migration to the RuOx complex to give diketones **20f** were identified. Based on the experimental data, it was concluded that the preferred pathway in oxidation was strongly dictated by the choice of reaction solvent. Pola et al. [79] synthesized [Zn(TPTTP)]Cl$_2$ **20a** and [Ru(TPTTP)]Cl$_2$ **20a′** complexes and explored their properties and utility. Both [Zn(TPTTP)]Cl$_2$ and [Ru(TPTTP)]Cl$_2$ complexes were tested for their C–H oxidation of pyrene **2a**. Treatment of the Zn(II) or Ru(II) complexes with pyrene **2a** resulted in the formation of dione **20b** via the photo-oxidation pathway (Scheme 13).

Sequence of oxidation based on $^1$H NMR of monophasic, HPLC-MS analysis of biphasic reaction and DFT calculations (Nowicka and Willock *et al.*)

**Scheme 13.** Functionalization of the C9 and C10 positions of pyrenes. C–H oxidation of pyrenes [78,79].

### 2.5. C–H Functionalization and Involving the Pyrene Backbone

Mastarlerz and co-workers demonstrated [80] a new synthetic route to synthesize tetraindenopyrene **21f** using the C–H functionalization route. The synthesis of tetrain­denopyrene was accomplished starting with commercially available hexahydropyrene. Hexahydropyrene was selectively four-fold C–H brominated to afford **21b** with a 84 % yield via simple filtration. Subsequent Suzuki–Miyaura cross-coupling under Fu's conditions (Pd$_2$dba$_3$ and HP*t*-Bu$_3$BF$_4$) and oxidation of the unsaturated propylene tethers using DDQ gave the pyrene derivative **21d** with a 70% yield over two steps. The next step was the tetra C–H chlorination of pyrene **21d**, which was performed using a slight excess (4.5 equiv) of *N*-chlorosuccinimide (NCS) in chloroform to afford **21e** with a 94 % yield. Finally, the

synthesis of tetraindenopyrene **21f** was accomplished through the C–H activation step (Scheme 14).

**Scheme 14.** Synthesis of tetraindenopyrene via the C–H functionalization reaction and involving the pyrene backbone [80].

### 3. Directing Group-Assisted C–H Activation and Functionalization of Pyrenes, Affording Functionalized Pyrenes

In this section, we have presented the recent developments pertaining to the C–H activation and functionalization of pyrene motifs using directing groups. There have been successful efforts in using the direct C–H activation/functionalization of different positions of the pyrene core (as described in the previous section). The efficiency and region- or site-selectivity issues and expansion of the scope of C–H functionalization of pyrene were tackled using the directing group-assisted C–H activation concept.

### 3.1. Directing Group-Assisted C–H Functionalization of the C10 Position of Pyrenes

Miura et al. revealed [81] two examples of copper-mediated dehydrogenative biaryl coupling of the pyrene core using the picolinamide bidentate directing group-assisted C–H functionalization strategy [81–89]. Substrate *N*-(pyren-1-yl)picolinamide **22a** was treated with the azole derivatives **22aa** and **22ab** (2 equiv) in the presence of Cu(OAc)$_2$ (3 equiv) and PivOH (1 equiv) in mesitylene at 165 °C for 4 h under a nitrogen atmosphere (Scheme 15). These trials successfully afforded the C10-functionalized pyrene derivatives **22ca** and **22cb** of 71% and 69% yields, respectively. A plausible mechanism [81] for the C10 site-selective functionalization process involving the bidentate directing group, the chelation-assisted pathway, has been proposed. The C–H cupration of a relatively acidic C–H bond of **22ab**, followed by *N,N*-bidentate coordination with **22a,** generates the organocopper intermediate **22d**. Subsequently,

substrate **22a** undergoes C–H cleavage, accompanied by oxidation with additional Cu(II), to form the Cu(III) metallacycle **22f**. Reductive elimination then generates the corresponding C10-functionalized pyrene **22cb**. The site-selectivity of this reaction is guided by the formation of the kinetically favored five-membered metallacycle intermediate **22e**.

**Scheme 15.** Functionalization of the C10 position of pyrenes. Copper-mediated dehydrogenative biaryl coupling of 1-aminopyrene with 1, 3-azoles [81].

Our research group [89] reported the application of the bidentate directing group-assisted C–H functionalization tactics to functionalize the pyrene core. The relatively inaccessible K-region C10 position of the pyrene core was subjected to C–H arylation and alkylation. The Pd(II)-catalyzed γ-C–H arylation and alkylation of the C10 position of *N*-(pyren-1-yl)picolinamide possessing a picolinamide bidentate directing group resulted in various C1- and C10-disubstituted pyrene scaffolds (**23a,b**) (Scheme 16). We have also shown the removal of the picolinamide directing group following the C–H arylation/alkylation reactions. The structures of representative pyrene derivatives were confirmed via X-ray structure analysis, which confirmed the site-selective γ-C–H functionalization of the inaccessible K-region C10 position of the pyrene core. Given the importance of the pyrene derivatives across different fields of chemical sciences, our work contributed to the augmentation of the library of pyrene derivatives with C1- and C10-disubstituted pyrene amide motifs [89].

**Scheme 16.** Functionalization of the C10 position of pyrenes. Pd(II)-catalyzed directing group-aided C–H arylation and alkylation of the pyrene core. Synthesis of C1- and C10-disubstituted pyrene scaffolds [89].

Nishihara et al. reported [90] an example of *peri*-selective chalcogenation of 1-aminopyrene with diaryl disulfide through C–H bond cleavage using a palladium catalyst (Scheme 17). *N*-(pyren-1-yl)picolinamide **22a**, diphenyl disulfide (1.2 equiv), PdCl$_2$(NCPh)$_2$ (10 mol%), CuCl$_2$ (10 mol%), and PivOH (1.2 equiv) in DMSO were heated at 100 °C for 12 h to afford *N*-(10-(phenylthio)pyren-1-yl)picolinamide with a 62% yield. A plausible mechanism for the C10 site-selective C–H chalcogenation reaction involving the bidentate directing group, the chelation-assisted pathway, has been proposed [90]. Palladium complex **24c** is formed with picolinamide; then, cyclopalladation selectively occurs at the C10 position of the pyrene core to afford **24d**. Oxidative addition of diphenyl disulfide to **24d** generates **24e**. Reductive C-S bond formation from **24e** gives the product ligand complex **24f**. Ligand exchange in **24f** affords **24g**. The final product **24b** dissociates from **24g**, while **22a** simultaneously captures the released palladium in the catalytic cycles. PivOH may assist in the cleavage of the C–H bond in the concerted metalation-deprotonation (CMD) step to generate **24d**.

**Scheme 17.** Functionalization of the C10 position of pyrenes. Pd-catalyzed *peri*-selective C–H chalcogenation of 1-aminopyrene with diaryl disulfide and Cu(II)-mediated C–N coupling involving pyrenes and indoles [90,91].

In 2017, Punniyamurthy et al. demonstrated [91] a copper(II)-mediated chelation-assisted regioselective C–N coupling involving pyrene and indoles through the dehydrogenative cross-coupling method and an example of *N*-pyrenylation of a 5-methoxy indole affording **24i** (Scheme 17). The *N*-(pyren-1-yl)picolinamide **22a** was treated with 5-methoxy-1*H*-indole (2 equiv) in the presence of Cu(OAc)₂ (20 mol%), Ag₂CO₃ (25 mol%), K₃PO₄ (2 equiv), and NMO (2 equiv) in DMSO at 140 °C for 14 h, and this reaction gave the C10-functionalized pyrene **24i** with a 35% yield. The plausible mechanism for the site-selective C–N coupling reaction involving pyrene and indole affording **24i** has been

proposed. Initially, *N*-cupration of substrate **22a** with Cu(OAc)₂ may form **24j**, which may undergo substitution with indole **24h** to generate the intermediate **24k**. Then, the intermediate **24k** may generate **24l** in the presence of Cu(OAc)₂ via oxidative addition, which can lead to pyrenyl C($\gamma$)–H cupration to make organocopper(III) species **24m**. Then, the C–N coupled product **24i** may be formed through reductive elimination from **24m**.

Punniyamurthy et al. unveiled [92] a copper-mediated regioselective C–H etherification of 1-aminopyrene with arylboronic acids utilizing water as an oxygen source (Scheme 18). The *N*-(pyren-1-yl)picolinamide substrate **22a**, phenylboronic acid (2 equiv), Cu(OAc)₂ (1.5 equiv), Cs₂CO₃ (2.5 equiv), and DMSO mixture was heated at 130 °C for 8 h, which resulted in an etherification product **25a**. The etherification reaction occurred at the C10 position of pyrene core **22a**. A plausible mechanism for the selective C10 etherification of pyrene was proposed. Initially, the reaction of Cu(OAc)₂ with substrate **22a** in the presence of a base forms a Cu(II) species, which is further oxidized by Cu(OAc)₂, leading to the formation of a tetracoordinated Cu(III) complex **25b**. Next, an intramolecular cyclometallation process through C–H bond activation forms a copper(III) species **25c**. This copper(III) species can then react with a boronate complex and water, forming a copper(III) intermediate **25d**. The reductive elimination step then affords the etherification product **25a**.

**Scheme 18.** Functionalization of the C10 position of pyrenes. Copper-mediated regioselective C–H etherification of 1-aminopyrene with arylboronic acid using water as an oxygen source [92].

Chatani's group reported [93] a rhodium(I)-catalyzed C(10) alkylation of 1-aminopyrene with alkenes through a bidentate picolinamide chelation system, affording **26b** (Scheme 19). A plausible mechanism for the rhodium(I)-catalyzed C(10) alkylation of 1-aminopyrene with alkenes through a bidentate picolinamide chelation system, affording **26b**, has been proposed. Initially, the amide N–H bond of pyrene substrate **22a** undergoes oxidative addition to the

Rh center, leading to the formation of a Rh(III) hydride species **26c**. Then, intermediate **26c** eliminates a carboxylic acid motif, resulting in the generation of species **26h**. It is also possible that the species **26c** can form intermediate **26i**. The hydride species **26c** can reversibly bind to the acrylate motif. The acrylate then undergoes insertion into the H–Rh bond in intermediate **26d**, generating the species **26e**. Then, species **26e** releases a carboxylic acid motif, leading to the formation of the carbine species **26f**. Subsequently, a C–H insertion into species **26f** occurs, generating the cyclometalated species **26g**. Finally, in the presence of carboxylic acid, reductive elimination takes place, affording the C10-alkylated pyrene, and the active Rh(I) catalyst is regenerated.

**Scheme 19.** Functionalization of the C10 position of pyrenes. Rhodium(I)-catalyzed C(10) alkylation of 1-aminopyrene with alkenes through a bidentate picolinamide chelation system [93].

Wu et al. described [94] the cobalt-catalyzed C–H carbonylation of 1-aminopyrene and synthesis of pyrene-derived (*NH*)-benzo[*cd*]indol-2(1*H*)-one through a bidentate picolinamide chelation system (Scheme 20). This reaction employs picolinamide as a traceless directing group and uses benzene-1,3,5-triyl triformate (TFBen, **27a**) as the CO source,

affording **27b**. A plausible mechanism for the cobalt-catalyzed C–H carbonylation of 1-aminopyrene and synthesis of **27b** through a bidentate picolinamide chelation system has been proposed. Initially, the Co(II) catalyst coordinates with **22a** and is oxidized by Ag(I) to generate the Co(III) complex **27c**, which undergoes a selective C–H activation reaction at the C10 position of **22a**, leading to the formation of intermediate **27d**. Then, coordination of in situ-generated CO from TFBen gives the acyl Co(III) complex **27e**, which undergoes reductive elimination, generating species **27f**. The hydrolysis of species **27f** gives the expected product **27b** and releases Co(I) species. The active Co(II) catalyst is then regenerated through the oxidation of Co(I) by Ag(I) in the catalytic cycle.

**Scheme 20.** Functionalization of the C10 position of pyrenes. Cobalt-catalyzed C–H carbonylation of 1-aminopyrene and synthesis of the pyrene-derived (*NH*)-benzo[*cd*]indol-2(1*H*)-one derivative **27b** through a bidentate picolinamide chelation system [94].

Feng et al. revealed [95] an example of cobalt-catalyzed hydroarylation of 1,3-diynes with *N*-(pyren-1-yl)picolinamide promoted by TFE, affording C10-alkenylated pyrene **28b**

(Scheme 21). Substrate *N*-(pyren-1-yl)picolinamide, 1,4-diphenylbuta-1,3-diyne (2 equiv), Co(OAc)$_2$·H$_2$O (30 mol%), and KOAc (2 equiv) in TFE were heated at 100 °C for 12 h, which successfully gave (*E*)-*N*-(10-(1,4-diphenylbut-1-en-3-yn-1-yl)pyren-1-yl)picolinamide (**28b**) with a 76% yield. A plausible mechanism for the cobalt-catalyzed hydroarylation of 1,3-diynes with *N*-(pyren-1-yl)picolinamide promoted by TFE, affording C10-alkenylated pyrene **28b**, has been proposed via intermediates **28c** and **28d**. The protonolysis of intermediate **28d** results in the formation of **28b**, and the protonolysis step is facilitated by trifluoroethanol.

**Scheme 21.** Functionalization of the C10 position of pyrenes. Cobalt-catalyzed hydroarylation of 1,3-diynes with *N*-(pyren-1-yl)picolinamide promoted by TFE, affording C10-alkenylated pyrene **28b** [95].

Punniyamurthy showed [96] an example of copper-mediated oxidative C–H/N–H annulation of 1-aminopyrene with diethyl malonate, affording **29b** through a bidentate picolinamide chelation system (Scheme 22). Substrate *N*-(pyren-1-yl)picolinamide **22a** was treated with diethyl malonate (2 equiv) in the presence of Cu(OAc)$_2$ (1 equiv) and NaOPiv·H$_2$O (2 equiv) in dimethyl sulfoxide at 120 °C for 4 h in air. This reaction gave the diethyl 3-picolinoylcyclopenta[*cd*]pyrene-4,4(3*H*)-dicarboxylate product **29b** with a 51% yield. A plausible mechanism for the copper-mediated oxidative C–H/N–H annulation of 1-aminopyrene with diethyl malonate, affording **29b** through a bidentate picolinamide chelation system, has been proposed. First, diethyl malonate **29a** reacts with Cu(OAc)$_2$ in the presence of a base to give intermediate **29c**, which reacts with *N*-(pyren-1-yl)picolinamide **22a** and forms species **29d**. Then, **29d** may oxidize to generate species **29e** in the presence of Cu(OAc)$_2$ and can further undergo *ortho*-C(sp$^2$)–H cupration to deliver the organocopper(III) complex **29f**. Species **29f** then affords **29g** via reductive elimination. Product **29b** is then formed via intramolecular N–H/C(sp$^3$)-H dehydrogenative cross-coupling in the presence of Cu(OAc)$_2$ and bases.

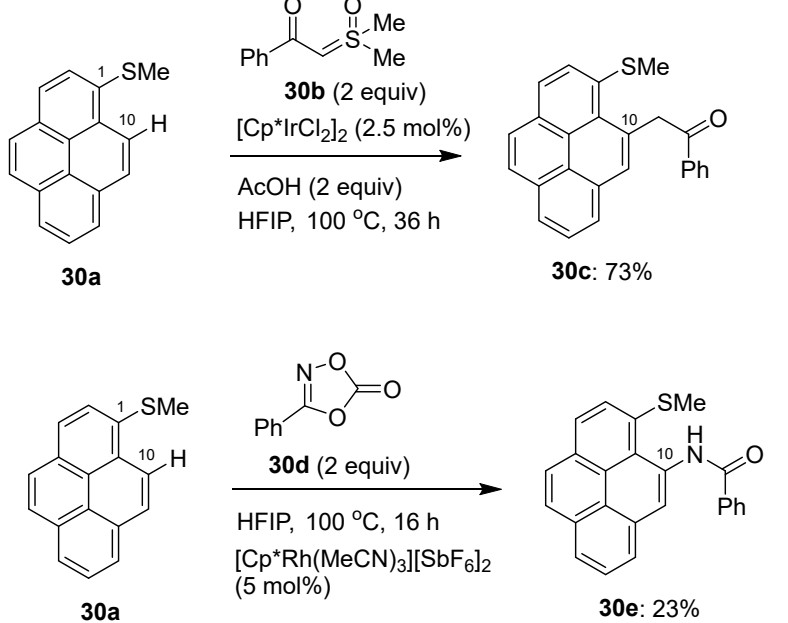

**Scheme 22.** Functionalization of the C10 position of pyrenes. Copper-mediated oxidative C–H/N-H annulation of 1-aminopyrene with diethyl malonate, affording **29b** [96].

Miura et al. showed [97] an iridium-catalyzed acylmethylation reaction and a rhodium-catalyzed amidation reaction at the C10 position of a pyrene moiety via the thio group-assisted C–H activation strategy. The reaction of **30a** with sulfoxonium ylide **30b** in the presence of [Cp*IrCl₂]₂ (2.5 mol%) and AcOH (2 equiv) in HFIP at 100 °C afforded the C10-alkylated pyrene moiety **30c** with a 73% yield. Similarly, the reaction of **30a** with the dioxazolone moiety **30d** in the presence of [Cp*Rh(MeCN)₃][SbF₆] (5 mol%) in HFIP at 100 °C gave the C(10) amidation product **30e** with a 23% yield (Scheme 23).

**Scheme 23.** Functionalization of the C10 position of pyrenes. *peri*-selective direct acylmethylation and amidation of pyrene using iridium and rhodium catalysts [97].

Hierso and Roger reported [98] the Rh(I)-catalyzed, diphenylphospino group-assisted C(10)–H arylation of 1-pyrenylphosphine. A wide range of *ortho-*, *meta-*, and *para-*substituted aryl bromides and heteroaryl bromides were reacted with 1-pyrenylphosphine in the presence of the [Rh(COD)$_2$)]BF$_4$ catalyst to afford the corresponding C(10)-arylated 1-pyrenylphosphine derivatives (Scheme 24). Notably, 1-pyrenylphosphine was also reacted with bulky *ortho-*functionalized bromoarenes 1- or 2-bromonaphthalene, 9-bromophenanthrene, 9-bromoanthracene, 3-bromofluoranthene, and 1-bromopyrene to afford the corresponding 1-pyrenylphosphine derivatives. The observed selective arylation at the C(10) position of the pyrene core was confirmed with X-ray structures of representative C(10)-arylated1-pyrenylphosphine derivatives. The C(10)–H arylation reaction of 1-pyrenylphosphine was also accomplished using a Rh(III) catalyst (e.g., [RhCl$_2$(Cp*)]$_2$). This encouraged the authors to synthesize the Rh(III) complexes **31aa**, **31ab**, and **31ac,** which are expected to be the key intermediate complexes in the catalytic process affording the C(10)-arylated 1-pyrenylphosphine (Scheme 25). Complex **31aa** was synthesized from 1-pyrenylphosphine and [RhCl$_2$(Cp*)]$_2$. Next, complex **31ab** was synthesized from 1-pyrenylphosphine, [RhCl$_2$(Cp*)]$_2$ and KOAc. The structure of complexes **31aa** and **31ab** was confirmed via X-ray structure analysis. Then, the Rh(III) cationic complex **31ac** was obtained by treating the complex **31ab** with 1-pyrenylphosphine in the presence of KPF$_6$. Then, complexes **31aa**, **31ab**, and **31ac** were employed as catalysts in the C(10)–H arylation of 1-pyrenylphosphine with 4-bromoanisole. The Rh(III) complex **31aa** and the cationic metallacycle **31ac** gave the C(10)-arylated 1-pyrenylphosphine **31ai** of 77–88% yields, and the metallacycle **31ab** gave the C(10)-arylated 1-pyrenylphosphine **31ai** of a low yield (29%). It was found that the metallacycle **31ab** may travel out of the catalytic cycle for a while by forming complex **31ad** through halogen exchange. A plausible mechanism for the Rh-catalyzed C(10)–H arylation of 1-pyrenylphosphine was stated (Scheme 25) [98]. An ortho-metalation reaction occurs in complex **31aa** to afford **31ab** through a concerted metalation-deprotonation reaction assisted by the base, which then generates the cationic complex **31ac**. Then, the oxidative addition of aryl bromide on the complex **31ac** gives the complex **31ae** through a reductive elimination/C–C bond formation process. Subsequently, the coupling product **31b** is released, generating the cationic complex **31af** that is ready for further cyclometallation process in the catalytic cycle. Overall, Hierso and Roger's report described a facile condition for the peri-functionalization of the topical π-extended 1-pyrenylphosphine derivative using a cationic Rh(III) C–H arylation strategy. The arylation reaction took place at the K-region of the pyrene core, which is generally inaccessible by conventional organic synthesis.

### 3.2. Directing Group-Assisted C–H Functionalization of the C2 and C7 Positions of Pyrenes

Nakamura et al. described [99] an example of iron-catalyzed C2 allylation of pyrene-1-carboxamide with allyl phenyl ether **32a**, affording **32b** through a bidentate 8-aminoquinoline system (Scheme 26). Substrate *N*-(quinolin-8-yl)pyrene-1-carboxamide **22b**, allyl phenyl ether (2 equiv), Fe(acac)$_3$ (20 mol%), dppen (10 mol%), ZnCl$_2$·TMEDA (2 equiv), and *t*-BuCH$_2$MgBr (3.4 equiv) in THF were heated at 70 °C for 135 h to afford the 2-allyl-*N*-(quinolin-8-yl)pyrene-1-carboxamide **32b**. Nakamura et al. noticed [99] an unexpected generation of intermediate **32d** that formed after the cleavage of the C–H bond of the substrate under investigation, which interacts with an allyl ether to afford **32e** via path A. This is in contrast to the conventional cross-coupling reaction involving an allyl group (path B) or the extensively studied oxidative C–C bond formation process (path C), both of which have been extensively explored (Scheme 26).

**Scheme 24.** Functionalization of the C10 position of pyrenes. Phosphorus-directed rhodium-catalyzed C–H arylation of 1-pyrenylphosphines is selective at the K-region [98].

**Scheme 25.** Functionalization of the C10 position of pyrenes. Phosphorus-directed rhodium-catalyzed C–H arylation of 1-pyrenylphosphines selective at the K-region [98].

**Scheme 26.** Functionalization of the C2 position of pyrenes. Iron-catalyzed C2-allylation of pyrene-1-carboxamide with allyl phenyl ether and Pd-catalyzed C2-arylation of pyrene-1-carboxamide with the help of the 8-aminoquinoline bidentate directing group [89,99].

Our research group [89] also reported the application of the 8-aminoquinoline bidentate directing group-assisted C–H functionalization tactics to functionalize pyrene-1-carboxamide. The relatively inaccessible C2 position of the pyrene core **22b** was subjected to C–H arylation and alkylation. The Pd(II)-catalyzed $\beta$-C–H arylation and alkylation of the C2 position of pyrene-1-carboxamide **22b** possessing the 8-aminoquinoline bidentate directing group yielded various C1- and C2-disubstituted pyrene scaffolds **32g,h** (Scheme 26). The structures of representative pyrene derivatives were confirmed via X-ray structure analysis, which confirmed the site-selective C–H functionalization of the inaccessible C2 position of the pyrene core. This work contributed to the augmentation of the library of pyrene derivatives with C1- and C2-disubstituted pyrene amide motifs [89].

In 2020, Larrosa and co-workers [100] successfully synthesized a diverse range of C–H-arylated pyrene derivatives (**33b**) by employing a palladium-catalyzed C–H *ortho*-arylation method on pyrene-1-carboxylic acid (**33a**) (Scheme 27). This approach provided a convenient route to access a variety of arylated pyrene compounds with substituents at the C1- and C2-positions. The C1 substituent (carboxylic acid unit) was converted into different functional groups, such as iodide, alkynyl, aryl, or alkyl functionalities. By utilizing this flexibility, the Larrosa group was able to produce arylated pyrene ammonium salts, which exhibited superior performance compared to the original non-arylated compound in the aqueous liquid phase exfoliation (LPE) process of graphite. Using the PEPPSI-IPr catalyst, the decarboxylative C–H arylation of pyrene-1-carboxylic acid **33a** gave C2-arylated pyrenes (**33c**). Pyrene-1-carboxylic acid (**33a**) was treated with aryl iodides (3 equiv) in the presence of Pd(OAc)$_2$ (6 mol%), KOAc (2.8 equiv), NMe$_4$Cl (2.16 equiv), and AcOH (1.5 equiv) at 120 °C for 45 h, resulting in the formation of C–H-arylated pyrene derivatives (**33b**) with up to a 90% yield. The arylated pyrene-1-carboxylic acids (**33b**) were subsequently converted into their corresponding iodoarene **33d** by treating pyrene-1-carboxylic acid (**33b**) with I$_2$ (3 equiv) and K$_3$PO$_4$ (1 equiv) in *ortho*-dichlorobenzene (0.2 M) at 120 °C for 21 h. On the other hand, 2-arylpyrene (**33c**) was synthesized by treating pyrene-1-carboxylic acid (**33a**) with aryl iodide (3 equiv) in the presence of PEPPSI-Ipr (2 mol%), Ag$_2$CO$_3$ (1 equiv) and AcOH (1 M) at 150 °C for 19 h. Iodopyrene (**33d**) underwent further derivatization through well-stabilized cross-coupling reactions, such as the Suzuki and Sonogashira reactions using arylboronic acid, primary aryl alkynes, and trimethylsilylacetylene. These reactions resulted in the formation of the corresponding products **33e** and **33f**. The Pd-catalyzed C–H arylation of pyrene-1-carboxylic acid is believed to occur through a Pd(II)-Pd(IV) mechanism. Initially, cyclometallation takes place, leading to the formation of **33g**, following which a Pd(IV) intermediate (**33h**) is generated upon oxidative addition of ArI with **33g**. Subsequently, rapid reductive elimination occurs, resulting in the production of the arylated carboxylic acid **33b** and the restoration of Pd(II) species.

Zhong et al. reported [101] a Ru(II)-catalyzed pyridyl moiety-directed protocol for the synthesis of C2- or C2- and C7-arylated pyrene derivatives via the C–H activation pathway under microwave and conventional heating conditions (Scheme 28). The pyrene core, containing the pyridyl moiety, was treated with aryl bromides with [RuCl$_2$(*p*-cymene)]$_2$ (5 mol%), MesCOOH (0.6 equiv), and K$_2$CO$_3$ (4 equiv) in toluene under reflux for 36 h to afford the C–H arylated pyrenes **34b**/**35b** (C2- or C2- and C7-arylated pyrene derivatives). A plausible reaction pathway for the Ru(II)-catalyzed pyridyl moiety-directed C–H arylation of pyridine core **34a** was proposed (Scheme 29). The coordination of the pyridine moiety with the Ru catalyst, followed by a metalation-deprotonation C–H bond activation process at the C2 position, gave the intermediate **36c**. The C–H activation reaction is possibly promoted by carbonyl oxygen of the MesCOO$^-$ group, as represented in species **36b**. Due to the steric hindrance, the possibility of C–H activation at the C10 position of the pyrene core is neglected. Dissociation of the MesCOO$^-$ unit from species **36c**, followed by oxidative addition of aryl iodide with **36c**, gives the cationic Ru(IV) intermediate **36d**. Reductive elimination of species **36d** affords the C2-arylated pyrene derivative **34c** and the Ru catalyst.

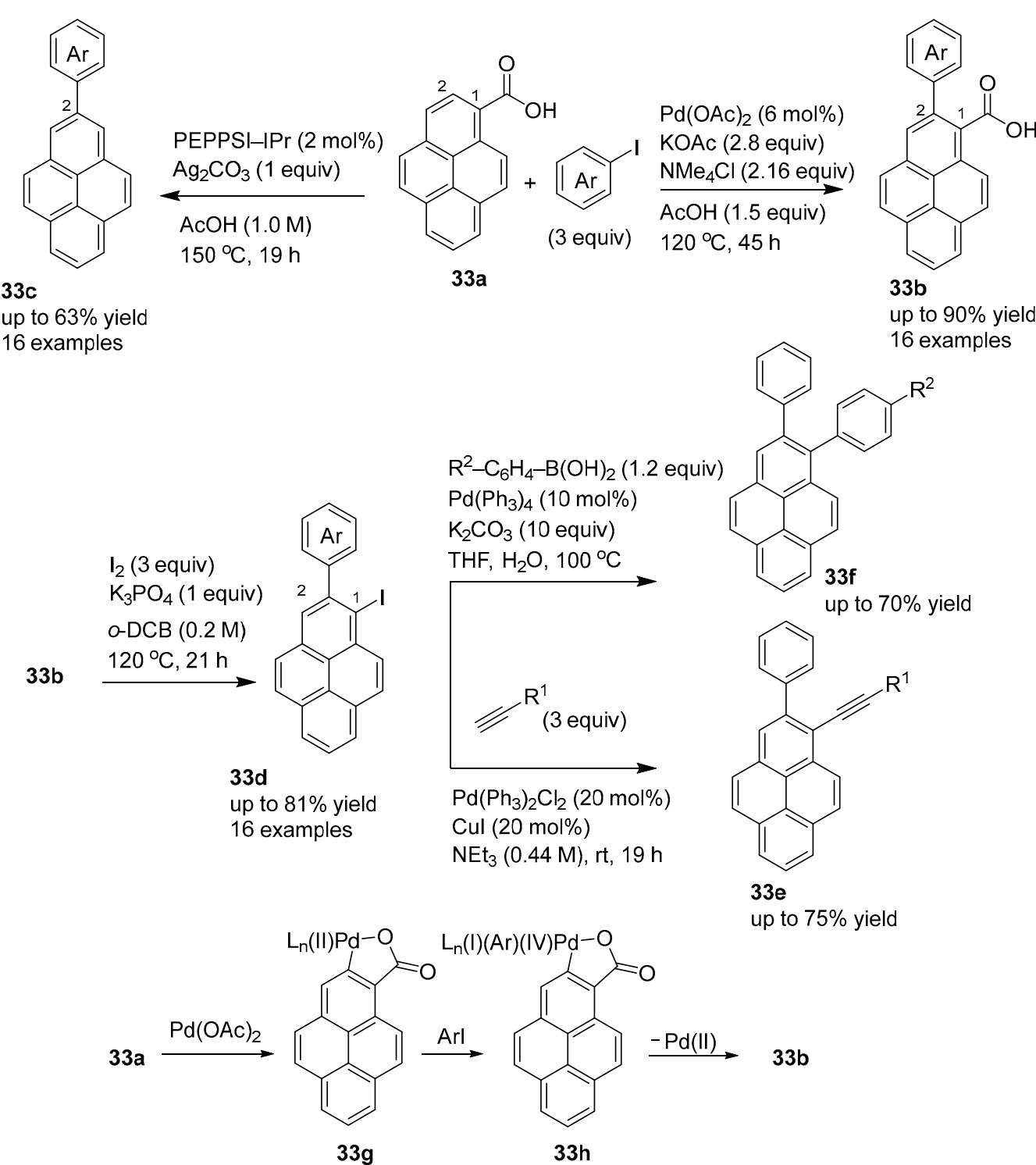

**Scheme 27.** Functionalization of the C2 position of pyrenes. Pd-catalyzed C–H arylation of pyrene-1-carboxylic acid [100].

**34ba**: R = OMe (92% / 91%)
**34bb**: R = H (85% / 88%)
**34bc**: R = Me (82% / 86%)
**34bd**: R = Cl (85% / 71%)
**34be**: R = CO$_2$Et (80% / 89%)
**34bf**: R = CF$_3$ (74% / 84%)

**34bg** 80% / 77%

**34bh** 77% / 85%

**35ba**: R = OMe (69% / 71%)
**35bb**: R = H (80% / 65%)
**35bc**: R = Me (77% / 65%)
**35bd**: R = Cl (63% / 52%)
**35be**: R = CO$_2$Et (70% / 64%)
**35bf**: R = CF$_3$ (75% / 65%)

**35bg** 60% / 44%

**34bh** 46% / 56%

Condition A:
Ar-Br (1.2 equiv), [RuCl$_2$($p$-cymene)]$_2$ (5 mol%), MesCOOH (0.6 equiv), K$_2$CO$_3$ (4 equiv), reflux, 36 h toluene.
Condition B:
Ar-Br (10 equiv), [RuCl$_2$($p$-cymene)]$_2$ (10 mol%), MesCOOH (1.2 equiv), K$_2$CO$_3$ (8 equiv), microwave heating, 30 min, NMP.
Condition C:
Ar-Br (2.4 equiv), [RuCl$_2$($p$-cymene)]$_2$ (10 mol%), MesCOOH (1.2 equiv), K$_2$CO$_3$ (8 equiv), reflux 36 h, toluene.
Condition D:
Ar-Br (20 equiv), [RuCl$_2$($p$-cymene)]$_2$ (20 mol%), MesCOOH (2.4 equiv), K$_2$CO$_3$ (16 equiv), microwave heating, 30 min in NMP.

**Scheme 28.** Functionalization of the C2 and C7 positions of pyrenes. The synthesis of C2- and C2 and C7-arylated pyrene derivatives through Ru(II)-catalyzed C–H activation and arylation [101].

**Scheme 29.** A plausible reaction pathway for the Ru(II)-catalyzed pyridyl moiety-directed C–H arylation of the pyrene core [101].

### 3.3. Directing Group-Assisted C–H Functionalization of the C9 Position of Pyrenes

Yang et al. disclosed [102] an example of P=O moiety-directed arylation at the C(9) position of pyrene core **37a** using **37b** and a CuI catalyst in DCE at 100 °C (Scheme 30). The desired product **37c** was obtained through a plausible six-membered metalacyclic intermediate **37d**, followed by aryl migration. You et al. revealed [103] an example of Cu-catalyzed C9 arylation of pyrene-1-carboxamide (**37e**) with mesityl(phenyl)iodonium triflate (**37f**) through a bidentate 8-aminoquinoline system, affording **37g** (Scheme 30). Substrate *N*-(*tert*-butyl)pyrene-1-carboxamide **37e** was treated with mesityl(phenyl)iodonium triflate in the presence of Cu(OTf)$_2$ (10 mol%) in 1,2-dichloroethane at 70 °C for 24 h. This reaction afforded the C(9)-arylated pyrene-1-carboxamide derivative **37g** of 65%. A plausible mechanism for the Cu-catalyzed C9 arylation of pyrene-1-carboxamide **37e** with mesityl(phenyl)iodonium triflate (**37f**) through a bidentate 8-aminoquinoline system was proposed. Initially, Cu(I) is generated by the reduction or disproportionation of Cu(II) species. Then, phenyliodinium salt oxidizes Cu(I) to a highly electrophilic Cu(III)-phenyl intermediate. The coordination of the carbonyl oxygen of pyrene **37e** to the Cu(III)-phenyl intermediate generates Cu(III) species **37h**, which undergoes an aryl transfer reaction via the Heck-like four-membered ring transition state to afford intermediate **37i** with Cu(III) and an aryl group-added pyrene system. Then, the breakdown of the C(10)–Cu bond generates Cu(I), and simultaneously, the OTf$^-$ anion abstracts the proton from the C(9) position, generating the desired C(9)-arylated pyrene derivative **37g**. Our group also showed [89] two examples of C(9) arylation of pyrene-1-carboxamide. Substrate **37e** was heated with aryl boronic acid (1.25 equiv) in the presence of Pd(OAc)$_2$ (10 mol%) and NFSI (1.25 equiv) in 1,2-DCE (2 mL) at 90 °C for 24 h, which afforded the C(9) arylation of pyrene-1-carboxamide **37j** and **37k** of 53% and 55% yields, respectively (Scheme 30).

**Scheme 30.** Functionalization of the C9 position of pyrenes. Copper-catalyzed arylation of pyrenes assisted by the P=O group [102] Cu-catalyzed arylation of pyrene-1-carboxamide with mesityl(phenyl)iodonium triflate and Pd-catalyzed arylation of pyrene-1-carboxamide with aryl-boronic acid [89,103].

## 4. Annulation of C–H Bonds of Pyrenes, Affording Pyrenes Appended with Additional Rings

In this section, we have presented the recent developments pertaining to the annulation of C–H bonds of pyrenes, affording pyrenes appended with additional rings. Hissler's group reported [104] transition metal-catalyzed synthesis of benzophosphole-fused pyrene (**38c**) and benzosilole-fused pyrene (**38e**) (Scheme 31). A phosphine moiety was introduced in the pyrene core via the Suzuki coupling reaction involving **38a**. Compound **38c** was synthesized through the C–H activation step involving Chatani's reaction condition [105]. Similarly, pyrene derivative **38d** was obtained through the Suzuki coupling reaction using bromoiodobenzene as the coupling partner, followed by lithiation and quenching with Me$_2$SiHCl of 1-(2-iodophenyl)pyrene. Compound **38e** was synthesized through the C–H activation step involving Takai's reaction condition [106].

**Scheme 31.** Functionalization of the C2 position of pyrenes. Synthesis of benzophosphole-fused pyrene (**38c**) and benzosilole-fused pyrene (**38e**) [104–106].

Huang's group disclosed a facile synthesis of highly fluorescent polycyclic compounds with a pyrene core appended with additional cyclic rings via aza-Michael addition, followed by double C–H activation in the presence of visible light (Scheme 32) [107]. The reaction of 1,8-pyrenedione with two equivalents of *N,N'*-dimethylethylenediamine (DED) generates **39c** under visible light. A plausible photochemical reaction pathway was stated by the authors [107]. The first step of the process affording **39b** is the aza-Michael addition reaction, resulting in **39c**. The presence of a *cis*-enone moiety allows for 1,4-addition and hydrogen transfer. The formation of the exited state **39ea** from **39c** is envisioned as the reactive intermediate responsible for the C–H bond activation reaction (path A), affording **39b**. Two facile [1,6]-hydrogen shifts generate the carbon-centered biradical intermediate **39eb** from **39ea**, which then undergoes oxidation to give the pyrene core-based polycyclic compound **39b** in the presence of air. Alternatively, the transformation of intermediate **39c** to **39d** and then to **39b** is an isomerization process enabled by UV light of 254 nm in the presence of a base (paths B and C).

**Scheme 32.** Functionalization of the C9 and C10 positions of pyrenes. Synthesis of pyrene core-based polycyclic compounds [107].

Jin et al. developed [108] a Pd-catalyzed cascade reaction of *N,N*-dialkyl-substituted *o*-alkynylaniline appended to a pyrene core to construct a pyrene derivative (**41b**) via peri-C–H annulation, affording **41b** (Scheme 33). Treatment of *N,N*-dialkyl-substituted *o*-alkynylaniline appended to pyrene **41a** with Pd(OAc)$_2$ (10 mol%), Cu(OPiv)$_2$ (1.5 equiv), CsOPiv (0.5 equiv), and AgBF$_4$ (0.5 equiv) in DMAC at 120 °C for 14 h afforded the annulated pyrene derivative **41b**. The plausible mechanism was outlined for this reaction. The activation of the triple bond of **41a** using the cationic Pd catalyst results in intramolecular 5-*endo* heteroannulation and indolium-Pd intermediates (**41c**). Subsequently, a peri-C–H bond activation of the pyrene core takes place via a pivalate-assisted concerted metalation

deprotonation pathway through transition state **41e**. Subsequently, the dealkylation of **41e** via the nucleophilic attack of a pivalate counter anion to one of the hexyl groups in **41e** generates palladacycle species **41f**. Reductive elimination produces the cyclopenta-fused pyrene derivative **41b**. The Pd(II) catalyst can be regenerated via the oxidation of Pd(0) with the help of a Cu(II) oxidant.

**Scheme 33.** Functionalization of the C10 position of pyrenes. Pd-catalyzed indolization/*peri*-C–H annulation/*N*-dealkylation cascade scaffold in the pyrene core [108].

Ravat's group reported [109] stereospecific synthesis of pyrene-fused [7]-helicene compounds **42a** and **42c**, connected via hexagonal and heptagonal rings (Scheme 34). Compounds **42a** and **42c** were synthesized via a one-pot Suzuki coupling between **5c**

and **43a,b**, followed by a C–H activation reaction and a two-step Suzuki–Scholl reaction, respectively. The stereospecific synthesis of pyrene-fused [7]-helicene compounds **42a** and **42c** was accomplished with complete retention of configuration, and the chiroptical properties were studied.

**Scheme 34.** Stereospecific synthesis of pyrene-fused [7]-helicenes connected via hexagonal and heptagonal rings [109].

Procter et al. established [110] a metal-free annulation method for the synthesis of pyrene core-based benzothiophenes through a two-fold C–H functionalization reaction of pyrene **2a** at the C(1) and C(2) positions (Scheme 35). A one-pot annulation of pyrene proceeded through an interrupted Pummerer reaction/[3,3]-sigmatropic rearrangement/cyclization sequence. It has been proposed that the first stage of this process relied on an intermolecular interrupted Pummerer reaction between pyrene and allyl sulfoxide to afford the sulfonium intermediate **44d** by activating allyl sulfoxide with triflic anhydride and subsequent trapping with pyrene. Upon heating the mixture, the desired [3,3]-sigmatropic rearrangement of intermediate **44d**, followed by spontaneous acid-promoted cyclization, generates sulfonium salt **44f** via **44e**. In the next stage, the addition of a nucleophilic base, NEt₃, to the same reaction pot converts sulfonium salt **44f** into the desired pyrene core-based 2,3-dihydrobenzothiophene product **44b**. The conversion of **2a** into **44b** was accomplished in one pot. Then, exposing the pyrene core-based 2,3-dihydrobenzothiophene product **44b** to 2,3-dichloro-5,6-dicyano-1,4-benzoquinone gave access to the pyrene core-based benzothiophene product **44c**. This process has given access to a material-oriented heteroaromatic system-appended pyrene unit.

**Scheme 35.** Functionalization of the C1 and C2 positions of pyrenes. Stereospecific synthesis of pyrene-fused [7]-helicenes connected via hexagonal and heptagonal rings [110].

## 5. Miscellaneous C–H Functionalization Transformations Involving Pyrenes

In this section, we have presented miscellaneous C–H functionalization transformations involving pyrenes, affording modified pyrenes. Jin's group reported an iridium-catalyzed [111] regioselective *ortho* C2–H bond activation of the pyrene core with the help of an imine moiety (Scheme 36). The reaction between **45a** and [Cp*IrCl$_2$]$_2$ in DCM at 50 °C in the presence of sodium acetate resulted in a half-sandwich iridium complex via the C(2)–H bond activation of the pyrene core **45a**, affording **45b**. A mixture of [Cp*RhCl$_2$]$_2$ (0.05 mmol), NaOAc (0.3 mmol), **45aa** (0.1 mmol), and DMAD (0.1 mmol) was stirred at 80 °C in 1,2-dichloroethane (DCE) for 8 h to give an annulated pyrene derivative (**45e**) with a 93% yield. A mixture of [Cp*IrCl$_2$]$_2$, NaOAc and **45aa** (0.1 mmol) gave **45be**, which, upon treatment with DMAD, gave the alkenylated species **45c**. Treatment of **45c** with NaBH$_4$ gave the alkenylated species **45d** with a 95% yield. Along this line, treatment of **45be** with a terminal alkyne resulted in an alkenylated species of **45f** with a 71% yield.

Ganguli et al. disclosed [112,113] a Ru(II)-catalyzed C–H activation reaction in both *ortho* and *peri* positions of the pyrene core, affording organometallic species **46b,c** (Scheme 37). The pyrene hydrazone motif **46a** was treated with different Ru catalysts and depending on the nature of the Ru(II) catalyst, the pyrene benzothiazole–hydrazone hybrid scaffold influenced both *ortho* and *peri* metalation via C–H activation of the pyrene core under an analogous reaction condition. The reaction of **46a** and RuHCl(CO)(PPh$_3$)$_3$ resulted in *peri* C–H metalation species, whereas the reaction of **46c** with RuH$_2$(CO)(PPh$_3$)$_3$ gave *ortho* metalation species. The C–H activation of **46a** using Wilkinson's catalyst at the *peri* position of the pyrene moiety was also reported by the same group. This reaction proceeded via the oxidative coordination pathway, where Rh(I) is oxidized to Rh(III), giving the C–H-activated product **46d,** and the benzothiazolylhydrazone moiety of **46a** acted as a directing group.

**Scheme 36.** Functionalization of the C2 position of pyrenes. Alkyne insertion induced regiospecific C–H activation with [Cp*MCl2]2 (M = Ir, Rh and Cp* = pentamethylcyclopentadienyl) [111].

**Scheme 37.** Functionalization of the C10 position of pyrenes. Organometallation in benzothiazolyl–hydrazone-derivatized pyrene [112,113].

## 6. Conclusions

In summary, in this review, we have shown some of the recent advances in the area pertaining to the modification of the pyrene core via the C–H activation and functionalization route. The 1-, 3-, 6-, and 8-positions of the pyrene motif were identified as 'active' or 'common sites'. The synthesis of functionalized pyrene derivatives by introducing substitutions at these sites is commonly explored. The 2- and 7-positions of pyrene are designated as 'nodal plane positions', and are considered 'uncommon' or 'less accessible sites for functionalization.' Other positions, namely 4-, 5-, 9-, and 10-, are called K-regions due to the carcinogenic effect of pyrene upon its oxidation. There have been significant efforts to functionalize the C–H bonds present in these regions. The site-selective C–H functionalization of the pyrene core was attempted with and without using directing groups. The C–H bonds present in the pyrene core were replaced with functional groups, and the corresponding functionalized and modified pyrenes were synthesized. The C–H functionalization method has enabled the introduction of functional groups with ease in the pyrene core and also allowed for the strengthening of the library of modified pyrene scaffolds. Various reviews describe the classical methods affording modified pyrenes. This review reported the developments in the area pertaining to the modification of the C–H bonds in the pyrene core. Pyrene and its derivatives have received significant attention in chemical sciences due to their superior fluorescence properties, efficient excimer emission, high charge carrier mobility, etc. Given this importance, we will witness newer protocols, including the C–H functionalization reaction for synthesizing functionalized pyrene derivatives.

**Author Contributions:** Conceptualization, S.A.B.; validation, S.A.B., A.D. and S.B.; data curation, S.A.B., A.D. and S.B.; writing—original draft preparation, S.A.B., A.D. and S.B.; writing—review and editing, S.A.B.; visualization, S.A.B.; supervision, S.A.B.; funding acquisition, S.A.B. All authors have read and agreed to the published version of the manuscript.

**Funding:** This research was funded by IISER Mohali to S.A.B. lab.

**Data Availability Statement:** Not applicable.

**Acknowledgments:** A.D. would like to thank IISER Mohali for providing the PhD fellowship. S.B. would like to thank CSIR, New Delhi, for providing the JRF and SRF fellowships.

**Conflicts of Interest:** The authors declare no conflict of interest.

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
