# Peer review of "Recent Advances in C–H Functionalization of Pyrenes"

_chemistry, doi:10.3390/chemistry5040175_

Round 1

Reviewer 1 Report

Comments and Suggestions for Authors

The review presented by Babu and co-workers focuses on the methods of site-selective functionalization of pyrene, most of which are based on metal-catalyzed C-H activation. Authors attempted to consider the important methods and problems which are related with the site-selective functionalization of pyrene and describe two different ways of C-H functionalization with and without directing group. In view that pyrene and its derivatives attract the attention of scientists as valuable objects to create new materials such as chemosensors, materials for bioimaging, organic electronics and other, the comprehensive review on this topic could be useful for readers. But this review requires significant revision and verification of the information provided. 

The following corrections are required:

1)     Priority requirement: it is necessary to carefully check the correspondence of the references to the information in the text. In many cases the references does not correspond to the text: see, for example, ref.24 (p.3);  ref.28 (p.4); ref.29a,b, 30 (p.6,7); ref. 27 is not mentioned in the text, etc.

Also, the phrase on p. 3: “In recent years, the C-H functionalization has emerged as one of the fruitful strategies to introduce a functional group at the C-H bonds of the small molecules and is considered an alternative method to the cross-coupling reactions” is accompanied by refs 25,26. Ref. 25: The review is published in Liebigs Ann. 1937 (!),531, 1-159; Ref. 26 is a short communication in J. Chem. Soc. Perkin 1. 1972, 1622-1623 on the synthesis of two compounds (1,6- and 1,8-dibromopyrenes). Thus this literature is too outdated.

2)     The numbering of compounds in the review should be carefully checked, because a large amount of the same compounds have the different numbers. For instance, 2e and 4a are the same compounds, 2f, 4b, 5e, 6b are also the same compounds, and 8 and 17c –again the same compounds, etc. Also, numbering is not required for simple starting compounds (for example, tBuCl (5a) in scheme 5; B2Pin2 (6a) in scheme 6; HSiEt3 (13a) in scheme 10, etc.

Also, letters or Roman numerals should be used for the intermediate species in the mechanistic schemes.

3)     Parts 2 and 3 should be reorganized according to the site of functionalization of pyrene, namely, in part 2 the literature data should be integrated in accordance with position of the reaction site in the pyrene fragment (for example, reaction at position 1, then at position 2 and then at position 10). The same systematization of data should be carried out in section 3 (for example, the approaches to the functionalization of C(10) pyrene position and then the approaches to the functionalization of C(2) pyrene position).

4)     In Scheme 2 the first reaction of C-H functionalization using an example of benzene is meaningless. I recommend combining this reaction with block “Modification of C-H bonds…”

5)     The review contains excessive information about the details of the experiment which should be removed (for example, lines 123-125, 136-139, 156-158, 191-194, 206-207, 226-228 etc)

6)     The references should be added to the captions of the schemes.

Comments on the Quality of English Language

Minor editing of English language required, for example, "Friedel-craft alkylation"

Author Response

Reviewer 1:

The review presented by Babu and co-workers focuses on the methods of site-selective functionalization of pyrene, most of which are based on metal-catalyzed C-H activation. Authors attempted to consider the important methods and problems which are related with the site-selective functionalization of pyrene and describe two different ways of C-H functionalization with and without directing group. In view that pyrene and its derivatives attract the attention of scientists as valuable objects to create new materials such as chemosensors, materials for bioimaging, organic electronics and other, the comprehensive review on this topic could be useful for readers. But this review requires significant revision and verification of the information provided

Our response: We thank the reviewer for giving valuable suggestions and recommending our article.

The following corrections are required:

1)     Priority requirement: it is necessary to carefully check the correspondence of the references to the information in the text. In many cases the references does not correspond to the text: see, for example, ref.24 (p.3);  ref.28 (p.4); ref.29a,b, 30 (p.6,7); ref. 27 is not mentioned in the text, etc.

Also, the phrase on p. 3: “In recent years, the C-H functionalization has emerged as one of the fruitful strategies to introduce a functional group at the C-H bonds of the small molecules and is considered an alternative method to the cross-coupling reactions” is accompanied by refs 25,26. Ref. 25: The review is published in Liebigs Ann1937 (!),531, 1-159; Ref. 26 is a short communication in J. Chem. SocPerkin 11972, 1622-1623 on the synthesis of two compounds (1,6- and 1,8-dibromopyrenes). Thus this literature is too outdated.

Our response: We thank the reviewer for pointing out these corrections. All these suggestions have been taken care of.

PS: We had submitted initial version without a template. Perhaps during processing by Editorial office according to template the mismatching of references seemed to occurred as earlier we had clubbed references, which were  singled out in order during processing.

2)     The numbering of compounds in the review should be carefully checked, because a large amount of the same compounds have the different numbers. For instance, 2e and 4a are the same compounds, 2f, 4b, 5e, 6b are also the same compounds, and 8 and 17c –again the same compounds, etc. Also, numbering is not required for simple starting compounds (for example, tBuCl (5a) in scheme 5; B2Pin2 (6a) in scheme 6; HSiEt3 (13a) in scheme 10, etc.

Also, letters or Roman numerals should be used for the intermediate species in the mechanistic schemes.

Our response: We thank the reviewer for pointing out these corrections. All these suggestions have been taken care of.

3)     Parts 2 and 3 should be reorganized according to the site of functionalization of pyrene, namely, in part 2 the literature data should be integrated in accordance with position of the reaction site in the pyrene fragment (for example, reaction at position 1, then at position 2 and then at position 10). The same systematization of data should be carried out in section 3 (for example, the approaches to the functionalization of C(10) pyrene position and then the approaches to the functionalization of C(2) pyrene position).

Our response: We thank the reviewer for pointing out these corrections. All these suggestions have been taken care of.

4)     In Scheme 2 the first reaction of C-H functionalization using an example of benzene is meaningless. I recommend combining this reaction with block “Modification of C-H bonds…”

Our response: We thank the reviewer for pointing out this correction. This suggestion has been taken care of.

5)     The review contains excessive information about the details of the experiment which should be removed (for example, lines 123-125, 136-139, 156-158, 191-194, 206-207, 226-228 etc)

Our response: We thank the reviewer for pointing out these corrections. We have curtailed the details and kept the relevant portions, and we request and would like to have some experimental details for the convenience of readers.

6)     The references should be added to the captions of the schemes.

Our response: We thank the reviewer for giving this suggestion. This suggestion has been taken care of.

Reviewer 2 Report

Comments and Suggestions for Authors

Pyrene and its derivatives have found applications in various branches of chemical sciences including organic chemistry, chemical biology, supramolecular and material sciences. Babu and coworkers have reported the recent developments in direct C-H activation and functionalization, directing group-assisted C-H activation and functionalization, annulation of C-H bonds of pyrenes. This review is well summarized. Therefore, I support the present manuscript for publication in Chemistry after considering the following comments.

(1) References to the literature should be numbered in one consecutive series according to the order they appear in the text. Many literatures in the references are not inconsistent with the description in the manuscript.

(2) In Scheme 13, “2” in the [Ru(TPTTP)Cl2] and [Zn(TPTTP)Cl2] should be subscript.

(3) In Scheme 31, the catalyst should be Pd(dppf)Cl2, not Pd(ddpf)Cl2.

(4) In Scheme 32, the “P” in Ph should be a capital letter.

Author Response

Reviewer 2:

Comments and Suggestions for Authors

Pyrene and its derivatives have found applications in various branches of chemical sciences including organic chemistry, chemical biology, supramolecular and material sciences. Babu and coworkers have reported the recent developments in direct C-H activation and functionalization, directing group-assisted C-H activation and functionalization, annulation of C-H bonds of pyrenes. This review is well summarized. Therefore, I support the present manuscript for publication in Chemistry after considering the following comments.

Our response: We thank the reviewer for giving valuable suggestions and recommending our article.

(1) References to the literature should be numbered in one consecutive series according to the order they appear in the text. Many literatures in the references are not inconsistent with the description in the manuscript.

Our response: We thank the reviewer for pointing out these corrections. This suggestion has been taken care of.

PS: We had submitted initial version without a template. Perhaps during processing by Editorial office according to template the mismatching of references seemed to occurred as earlier we had clubbed references, which were  singled out in order during processing.

(2) In Scheme 13, “2” in the [Ru(TPTTP)Cl2] and [Zn(TPTTP)Cl2] should be subscript.

Our response: We thank the reviewer for pointing out this correction. This suggestion has been taken care of.

(3) In Scheme 31, the catalyst should be Pd(dppf)Cl2, not Pd(ddpf)Cl2.

Our response: We thank the reviewer for pointing out this correction. This suggestion has been taken care of.

(4) In Scheme 32, the “P” in Ph should be a capital letter.

Our response: We thank the reviewer for pointing out this correction. This suggestion has been taken care of.

Round 2

Reviewer 1 Report

Comments and Suggestions for Authors

The authors significantly improved the Review, made all necessary corrections. The manuscript may be accepted in the present form for publication.